# The Role of Placental Non-Coding RNAs in Adverse Pregnancy Outcomes

**DOI:** 10.3390/ijms24055030

**Published:** 2023-03-06

**Authors:** Jiawen Ren, Heyue Jin, Yumin Zhu

**Affiliations:** 1Department of Maternal, Child and Adolescent Health, School of Public Health, Anhui Medical University, No 81 Meishan Road, Hefei 230032, China; 2MOE Key Laboratory of Population Health Across Life Cycle, School of Public Health, Anhui Medical University, No 81 Meishan Road, Hefei 230032, China; 3Anhui Provincial Key Laboratory of Population Health and Aristogenics, Anhui Medical University, No 81 Meishan Road, Hefei 230032, China; 4NHC Key Laboratory of Study on Abnormal Gametes and Reproductive Tract, Anhui Medical University, Hefei 230032, China

**Keywords:** placenta, miRNA, lncRNA, circRNA, adverse pregnancy outcomes

## Abstract

Non-coding RNAs (ncRNAs) are transcribed from the genome and do not encode proteins. In recent years, ncRNAs have attracted increasing attention as critical participants in gene regulation and disease pathogenesis. Different categories of ncRNAs, which mainly include microRNAs (miRNAs), long non-coding RNAs (lncRNAs), and circular RNAs (circRNAs), are involved in the progression of pregnancy, while abnormal expression of placental ncRNAs impacts the onset and development of adverse pregnancy outcomes (APOs). Therefore, we reviewed the current status of research on placental ncRNAs and APOs to further understand the regulatory mechanisms of placental ncRNAs, which provides a new perspective for treating and preventing related diseases.

## 1. Introduction

Adverse pregnancy outcomes (APOs) is a broadly defined term that includes various clinical outcomes such as preterm birth (PTB), miscarriage, macrosomia, low birth weight (LBW), birth defects, and others [1,2,3]. APOs are defined as the failure to give birth to offspring with a healthy appearance and function after pregnancy. Approximately 85% of women give birth at least once during their lifetime, whereas up to 30% of women may suffer from an APO [4,5,6]. At present, APOs are serious public health problems that have a significant impact on individuals, society, and finances worldwide [1,7,8].

The placenta plays an essential role in maintaining a healthy pregnancy, as well as aiding the growth and development of the fetus by providing a connection and barrier between mother and fetus, while abnormal alterations in the development and function of the placenta can cause a variety of pathological pregnancy conditions [9,10,11]. Placental formation and development involve complex networks of molecular regulation, and pregnancy-related diseases of placental origin are the result of a multifactorial combination of genetic and environmental influences. Therefore, we focused on the human placenta in this review. For many years, non-coding RNAs (ncRNAs) have been considered non-functional, while in recent years, the role of ncRNAs in post-transcriptional regulation has been discovered [12,13,14]. In addition, abnormal expression of ncRNAs is correlated with a variety of diseases, in which dysregulation of ncRNAs in the placenta is involved in various APOs (Figure 1).

Therefore, this review provides a summary of recent studies concerning placental ncRNAs associated with APOs, which mainly include miRNAs, lncRNAs, and circRNAs. Understanding their functions and regulatory mechanisms provides new insights into the prevention and treatment of related diseases.

## 2. Overview of the Class of ncRNAs

ncRNAs represent approximately 60% of the transcriptional production of the human genome [15,16], and there is a close association between many diseases and ncRNA mutations or abnormal expression [17,18]. ncRNAs can be divided into two categories based on their length: small ncRNAs and lncRNAs [19,20,21]. In addition, circRNAs represent a new class of ncRNAs that are derived from a process of “back-splicing” of pre-mRNAs with covalent binding between the downstream splice donor site and upstream splice acceptor site [22,23,24] (Figure 2).

miRNAs are a class of small ncRNAs (18–25 nt) that are endogenous, short, and highly conserved and are involved in a wide range of pathophysiological processes, including cell proliferation, growth, development, differentiation, and apoptosis [25,26,27]. The presence of miRNAs takes various forms: initially, miRNA exists as priRNA, which is processed into the precursor miRNA, called pre-miRNA. Then, pre-miRNA is cleaved by the Dicer enzyme and converted into mature miRNA [28,29,30]. miRNAs can bind to the target mRNAs, affecting the transcription and stability of the target mRNAs. In 1993, the identification of the first miRNA (*lin-4*) created the opportunity for miRNA research [31,32,33]. To date, more than 1000 different miRNAs have been identified in the human body, representing a powerful class of gene regulators [34].

Different from the short sequences of miRNAs, lncRNAs are usually longer than 200 nt, while some can be more than 100,000 nt and do not encode protein RNA transcripts [35,36]. The lncRNA primary transcripts were transcribed similarly to mRNA, including 5′ cap addition, 3′ poly (A), and splicing features. They are classified into five types according to their location in the genome: bidirectional lncRNA, intergenic lncRNA, sense lncRNA, antisense lncRNA, and intronic lncRNA [37,38,39]. There are increasing studies that have identified lncRNAs as a new class of regulatory molecules with the functions of scaffold, signal, and guide, and they are also involved in transcriptional interference [40,41,42].

circRNAs are a special type of endogenous ncRNAs [43,44]. Regarding circRNA formation, there are three main categories: exonic circRNA (ecircRNA), intronic circRNA (ciRNA), and exon-intron circRNA (EIcircRNA) [45]. circRNAs have various biological functions: (1) regulating the function of miRNAs by serving as a sponge for miRNAs; (2) acting as transcriptional regulators; (3) acting as protein translation vectors; and (4) regulating gene expression by interacting with proteins [46,47,48,49,50]. Moreover, it is obvious that circRNAs possess stable molecule properties, making them candidate markers for APOs [51,52].

## 3. Preterm Birth (PTB)

Preterm births, referring to births at less than 37 weeks of gestation, of which 80% are spontaneous PTBs (sPTBs), are a worldwide health problem and one of the causes of infant morbidity and mortality [53,54,55]. The majority of PTBs are idiopathic and spontaneous and not directly related to the medical causes of the event, such as pre-eclampsia [56,57]. Although several traditional risk factors for PTB were confirmed previously, including smoking, stress, infection, and family history, an understanding of the critical biological mechanisms and perturbed networks of PTB remains lacking [58,59,60]. Recently, there were several studies indicating that placental ncRNAs may play roles in the pathogenesis of PTB, mainly miRNAs and lncRNAs (Table 1).

### 3.1. miRNA and PTB

Placenta insufficiency is an important cause of PTB, and abnormal regulation of placental miRNAs can cause the onset of PTB. A lot of miRNAs can be detected in the trophoblast cells of the placenta. For instance, according to the study of Morales-Prieto et al. [73], 762 miRNAs were detected in trophoblast cells isolated from the placenta in the first and third trimesters of pregnancy. In addition, the majority of miRNAs were expressed as clusters in trophoblast cells, with the chromosome 14 miRNA cluster (C14MC) and chromosome 19 miRNA cluster (C19MC) being the most prominent, and these clusters appeared almost exclusively in trophoblast cells.

C19MC is mainly derived from trophoblast cells in late pregnancy and consists of at least 46 miRNAs [74]. According to recent studies, C19MC was discovered only in primates, with specific expression in the placental tissue [75,76,77]. Hromadnikova et al. [78] concluded that a characteristic phenomenon of PTB is having an upregulation of C19MC miRNAs by comparing the gene expression of C19MC miRNA in placental tissue between the following groups: full-term delivery, sPTB, and preterm premature rupture of membranes (PPROM). Tiozzo et al. [61] explored the expression of miRNA-519c, which is one of the types of C19MC, in placental tissue from PTB women and found that the level of miRNA-519c was significantly reduced in a preterm placenta with PPROM or chorioamnionitis than in a placenta delivered from spontaneous preterm delivery without PPROM or chorioamnionitis. Therefore, they speculated that the downregulation of miRNA-519c in the placentas may be linked to inflammation-associated PTB.

The above studies reveal the importance of miRNAs in PTB and provide new opportunities for discovering candidate biomarkers in the field of PTB, and future studies may consider validating these findings with larger sample sizes in different populations.

### 3.2. lncRNA and PTB

lncRNAs can regulate gene expression and are associated with PTB. Jiang et al. [62] found the overexpression of *SNHG29* in PTB placentas compared to full-term placentas, which activated P53/P21 signaling and triggered cellular senescence, causing PTB.

Premature rupture of membranes (PROM) and PTB are also inseparable. PPROM accounts for approximately one-third of all PTBs [79]. LUO et al. [80] used microarray analysis to identify 1954, 776, and 1050 differentially expressed lncRNAs in PPROM placentas in comparison with full-term birth (FTB), PTB, and PROM. Meanwhile, there were 449 and 3024 differentially expressed lncRNAs in PTB when compared with FTB and PROM groups, respectively. In addition, the study analyzed the metabolic pathways leading to PPROM and concluded that pathways of infection and inflammatory response, ECM-receptor interactions, apoptosis, actin cytoskeleton, and smooth muscle contraction are the major pathogenic mechanisms of PPROM. Additionally, the decreased expression of lncRNAs in the smooth muscle contraction pathway may increase placental smooth muscle contraction and induce PTB.

They further investigated the connection of the differentially expressed lncRNAs between sPTB and PPROM placentas and found an overlap at a coding locus, associated with the differential expression of transcribed mRNAs at the same locus [81]. Based on the finding that lncRNAs overlap with coding sites, this leads to the conclusion that differentially expressed lncRNAs in sPTB and PPROM human placentas probably regulate related mRNAs according to different mechanisms, but detailed mechanisms are yet to be clarified. Overall, these studies have opened a new avenue of exploration to understand the mechanism and function of lncRNAs in PTB. However, there are few reports on the relationship between the lncRNAs and PTB, which needs to be further studied.

## 4. Miscarriage

Miscarriages consist of spontaneous abortion (SA) and recurrent miscarriage (RM). SA, affecting approximately 10–15% of all pregnancies, is clinically defined as pregnancy loss before 28 weeks of gestation without any external intervention [82,83]. More than 80% of SAs are early SAs (within 12 weeks of gestation), and approximately 50–60% of early SAs are associated with chromosomal abnormalities of the embryo [84,85,86]. RM is characterized by two or more consecutive miscarriages with the same sexual partner before 28 weeks of gestation, with an incidence of 1% to 5%, and the risk of recurrence increases with the number of miscarriages [87,88,89]. Several recent studies have shown that ncRNAs participate in the regulation of miscarriage by affecting trophoblast cell proliferation, invasion, migration, and apoptosis (Table 1).

### 4.1. MiRNA and Miscarriage

Placental implantation and embryonic development are important aspects of pregnancy, and placental dysfunction is related to various complications during pregnancy, including miscarriage [90,91]. The trophectoderm constitutes the most important cellular component for placental implantation and maturation. There are numerous transcription factors, extracellular matrix factors, and certain adhesion molecules that strictly regulate human trophoblast cells. Recently, large miRNAs were detected in the human placenta, some of which may trigger miscarriage by regulating trophoblast function.

Tang et al. [92] performed a genome-wide screening of miRNAs to identify several miRNAs that were significantly differentially expressed in the chorionic villi tissue of patients with RM, with miRNA-4497 showing nearly 30-fold upregulated expression. They also investigated the potential targets of miRNA-4497 through software and databases to confirm the regulatory mechanism of miRNA-4497, finding that miRNA-4497 could target *SP1* mRNA directly; therefore, the overexpression of miRNA-4497 in the placenta of RM could downregulate the expression of *SP1*, which, in turn, induces trophoblast apoptosis and leads to RM [63]. A similar result was found in another study that increasing the expression of miR-27a-3p in the placental villus tissue of patients with RM downregulated ubiquitin-specific protease 25 (*USP25*), which contributes to the epithelial-to-mesenchymal transition (EMT) process, thus, suppressing the migration and invasion of trophoblast cells [65].

miRNAs also have a regulatory role in SA. For instance, Lu et al. [64] collected placental villi tissue from healthy pregnancies and unexplained SA cases for assay analysis, and miR-135a-5p was found to be significantly upregulated in the unexplained SA group. Next, the interaction of miR-135a-5p with Protein Tyrosine Phosphatase Non-Receptor Type 1 (*PTPN1*) was explored, concluding that miR-135a-5p likely promotes unexplained SA by targeting *PTPN1* to inhibit the proliferation, invasion, and migration of trophoblast cells.

These studies suggest that understanding the mechanisms of miRNAs that affect the trophoblast cells of the placenta contributes to elucidating the pathogenesis of miscarriage, as well as developing new strategies to diagnose and treat miscarriage early. Additionally, the presence of SNPs in miRNA genes alters the expression or maturation of miRNAs and is involved in the occurrence of miscarriage [93,94]. However, current studies have mainly focused on the exploration of miRNA SNPs in blood, whereas studies on placental miRNA SNPs are still lacking, which is pending future research in this area.

### 4.2. LncRNA and Miscarriage

Researchers have identified several lncRNAs that were differentially expressed in the placental tissue of patients with miscarriage compared to people without miscarriage. Xiang et al. [67] found that lncRNA *SNHG7-1* was downregulated in RSA placental villi compared to healthy pregnancy. In addition, *SNHG7-1* targets miR-34a through the Wnt/β-catenin signaling pathway to regulate the proliferation and invasion of trophoblast cells, thereby being involved in RSA.

Imprinted lncRNAs also have an important regulatory function during placental development. lncRNA *H19* is an imprinted gene that is paternally imprinted and maternally expressed and has a function of limiting embryonic development and slowing the growth rate of the offspring [95,96,97]. He et al. [66] found that lncRNA *H19* expression was significantly downregulated in the placental villi of patients with SA compared to women without SA but who terminated the pregnancy. They also found that *H19* inhibits the function of miRNA *let-7* to prevent mRNA degradation and thereby upregulates integrin β3 (*ITGB3*) expression, which indicates that it is involved in SA through the H19/let-7/ITGB3 axis. Sheng et al. [68] analyzed differentially expressed lncRNAs during mouse placental development through performing a microarray lncRNA screen and observed that the homologous sequence of imprinted lncRNA *Rian*, lncRNA *MEG8*, was found to be significantly upregulated in human SA villi, which inhibits the proliferation and invasion of trophoblast cells and is further involved in the development of unexplained SA. In summary, lncRNAs can induce the onset of miscarriage by regulating trophoblast function.

### 4.3. CircRNA and Miscarriage

circRNAs are likely to play a critical role in trophoblast cell invasion, EMT, apoptosis, and migration [98,99]. Recently, several researchers have begun to focus on the connection between circRNAs and SA. Zhu et al. [70] demonstrated that *circPUM1* may promote the processes and anti-inflammatory effects of trophoblast cells through the miR-30a-5p/JunB axis, which prevents the formation and development of SA. Additionally, Li et al. [69] demonstrated that circ-*ZUFSP* regulates trophoblast cell migration and invasion by the CIRC-ZUFSP/miR-203/STOX1 pathway in RSA. Another study showed that *circFOXP1* regulates trophoblast cell function through the miR-143-3p/S100A11 axis in placental tissue from patients with RSA [71].

Tang et al. [72] elucidated the expression of deregulated circRNAs and distinct competing endogenous RNA (ceRNA) networks by comparing unexplained recurrent spontaneous abortion (URSA) placental villi with those from healthy pregnancy by microarrays. They identified a unique circRNA, circRNA-0050703 (named circRNA-*DURSA*), that is downregulated in the placental villus of URSA. Furthermore, they found through the miR-760-HIST1H2BE axis that circRNA-*DURSA* could regulate apoptosis of trophoblast cells in URSA. There are few studies on circRNAs in miscarriage, and the specific mechanisms remain to be further explored.

## 5. Pre-Eclampsia (PE)

Hypertensive disorders of pregnancy (HDP) are pregnancy-specific disorders characterized by increased blood pressure during pregnancy, including gestational hypertension, pre-eclampsia, eclampsia, chronic hypertension with pre-eclampsia, and chronic hypertension [100,101,102]. The development of high-throughput sequencing techniques presents new ideas and methods for HDP research, and how placental ncRNAs participate in the pathological process of HDP has gained increasing attention, with PE being the most concerned. Placental ncRNAs can regulate placental development and the biological functions of trophoblast cells through their differential expression levels, which, in turn, affect the onset and progression of PE [103,104,105] (Table 2).

### 5.1. MiRNA and PE

Compared with placental tissue from healthy pregnancies, a population of differentially expressed miRNAs is present in the placenta with PE [126]. Using microarray methods, Brancaccio et al. [127] identified 298 differentially expressed miRNAs that are unique to the placenta of PE. Low-density lipoprotein receptor-associated protein 6 (*LRP6*) is a receptor for the Wnt/β-catenin signaling pathway, and its downregulation contributes to the development of PE. Zhou et al. [106] initially downloaded datasets from the Gene Expression Omnibus Database to evaluate miRNA candidates that regulate *LRP6* with validation in PE and healthy maternal placental villi tissue using RT-qPCR. They found that miR-513c-5p was upregulated in PE, which promoted trophoblast cell apoptosis and inhibited cell proliferation, invasion, and migration by directly targeting *LRP6* and downregulating its expression.

The expression of miR-21 was found to be upregulated in PE placental tissues and could directly bind to the 3’-UTR of forkhead box M1 (*FOXM1*) to downregulate the expression of *FOXM1*, thereby inhibiting trophoblast cell proliferation and participating in the development of PE [107]. Gunel et al. [108] demonstrated that the expression level of miR-195 was decreased in the placental tissues of PE patients and may participate in PE by regulating trophoblast cell proliferation, apoptosis, and angiogenesis. Another study found that the expression level of miR-145-5p was reduced in PE placental tissue and negatively regulated fms-related receptor tyrosine kinase 1 (*FLT1*), thereby inhibiting trophoblast cell proliferation and invasion [109].

Aberrant expression of insulin-like growth factor-1 (*IGF-1*) is present in placental tissues of PE patients. Niu et al. [110] found that miR-30a-3p has overexpressed in PE placental tissues with downregulated *IGF-1* expression, thereby affecting trophoblast apoptosis and invasion in order to participate in PE. Another study found that miR-548c-5p expression was significantly downregulated in the placental exosomes of PE patients and negatively regulated the expression of Recombinant Protein Tyrosine Phosphatase Receptor Type O (*PTPRO*) [111].

miR-210 was demonstrated to be expressed abnormally in the placental tissue of PE [128]. Placental hypoxia acts as a regulator for the occurrence and development of PE, and hypoxic culture can upregulate the expression of miRNA-210 in trophoblast cells, so miR-210 is closely related to PE [129]. Numerous other miRNAs are aberrantly expressed in PE placental tissue, such as miR-431, miR-518a-5p, and miR-124 being upregulated in PE placenta, while miR-3942, miR-532-5p, miR-423-5p, miR-127-3p, and miR-544 are downregulated [130,131]. To investigate the relationship between placental miRNAs and PE pathogenesis, the mechanisms of the placental miRNAs in the pathogenesis of PE will be further elucidated.

### 5.2. LncRNA and PE

Recently, lncRNAs were revealed to be involved in the development of PE by altering the biological functions of trophoblast cells. In the mechanism of lncRNA-miRNA interaction, lncRNAs act as ceRNAs for specific miRNAs, acting as miRNA sponge adsorbers and inhibiting the regulatory function of miRNAs, thereby regulating the expression of miRNA-targeted genes [132]. The expression of lncRNA *DLX6-AS1* is upregulated in PE placentas. lncRNA *DLX6-AS1* regulates the proliferation, invasion, and angiogenesis of trophoblast cells by acting as a sponge for miR-149-5p and influences the development of PE by affecting the expression of endoplasmic reticulum protein 44 (*ERP44*) [112]. The expression of lncRNA *XIST* is upregulated in the PE placenta, and *XIST* acts as a molecular sponge with miR-340-5p, which regulates the expression of potassium by inwardly rectifying channel subfamily J member 16 (*KCNJ16*), thereby inhibiting trophoblast cell proliferation and invasion and inducing cell apoptosis [113].

Yu et al. [114] demonstrated that the expression of lncRNA *SNHG16* was decreased in the placental tissues of PE patients, whereas low expression of lncRNA *SNHG16* inhibited trophoblast cell proliferation, migration, and invasion and induced apoptosis. The lncRNA *SNHG16* functions as a ceRNA for miR-218-5p and is involved in the development and progression of PE by promoting the expression of LIM and SH3 protein 1 (*LASP1*) to regulate the biological function of trophoblast cells. Another study showed that the expression of lncRNA *SNHG22* was significantly downregulated in PE placentas, with possible regulation of trophoblast cell migration and invasion by lncRNA *SNHG22* through binding to miR-128-3p and regulation of Protocadherin 11 X-Linked (*PCDH11X*) expression [115].

A critical lncRNA involved in PE progression is the lncRNA *TUG1*. The expression of lncRNA *TUG1* is downregulated in the PE placenta [116,117]. Li et al. [117] showed that lncRNA *TUG1* may promote trophoblast cell proliferation, invasion, and angiogenesis by targeting miR-29b to participate in the development of PE. Liu et al. [116] further demonstrated that lncRNA *TUG1* regulates the expression of vascular endothelial growth factor A (*VEGFA*) through binding to miR-29a-3p and activates the Ang2/Tie2 pathway, which promotes trophoblast cell proliferation, invasion, migration, and angiogenesis.

Moreover, the expression of lncRNAs *LINC00534* is upregulated in PE placental tissue [118], while the expression of lncRNAs *BCYRN1* is downregulated [119]. Expression changes in lncRNAs in the placenta affect the biological function of trophoblast cells through multiple pathways, which, in turn, affects placental function and is closely related to the occurrence and development of PE. Consequently, investigating PE-associated lncRNA expression can contribute to further elucidating the pathogenesis of PE.

### 5.3. CircRNA and PE

Research has demonstrated that circRNAs are involved in the occurrence and development of PE. Ma et al. [133] identified 361 differentially expressed circRNAs in PE placentas, of which 252 were upregulated and 109 were downregulated. Differential expression of circRNAs in PE placental tissues can influence trophoblast cells’ biological functions by indirectly regulating target genes and signaling pathways as miRNA sponges. Zhang et al. [120] identified that the expression level of *circSFXN* was increased in PE placentas and was involved in the development of PE by inhibiting trophoblast cell invasion and angiogenesis. The expression of *circZDHHC20* is upregulated in the PE placenta and *circZDHHC20* has a molecular sponge role with miR-144. It is involved in the development and progression of PE by promoting the expression of grainy head-like 2 (*GRHL2*) and, thus, regulating the biological functions of trophoblast cells [121].

Zhou et al. [123] reported that the expression level of *circPAPPA* was decreased in PE placental tissues. *circPAPPA* influences the development of PE by being a sponge of miR-384 to regulate trophoblast cell proliferation and invasion, together with the regulation of signal transducer and activator of transcription 3 (*STAT3*) expression. Another study by Li et al. [122] showed that *circPAPPA* can also regulate trophoblast cells via the miR-3127-5p/HOXA7 axis, causing PE.

The expression level of hsa_circ_0008726 was found to be elevated in PE placental tissue [124,125]. Shu et al. [124] demonstrated that hsa_circ_0008726 may act as a sponge for miR-345-3p and regulate the expression of RING1 and YY1 binding protein (*RYBP*), which, in turn, inhibits trophoblast cell migration, invasion, and EMT. Zhang et al. [125] revealed that hsa_circ_0008726 could be involved in PE by regulating the miR-1290-LHX6 pathway and inhibiting the proliferation, migration, and invasion of trophoblast cells. Numerous circRNAs were identified to be aberrantly expressed in PE placentas, but the potential functions and underlying mechanisms of these circRNAs in PE development have not been fully elucidated and need to be further investigated in the future.

## 6. Gestational Diabetes Mellitus (GDM)

Gestational diabetes mellitus refers to healthy glucose metabolism before pregnancy but varying degrees of abnormal glucose metabolism during pregnancy, which is a public health problem related to metabolic disorders that affect approximately 9–15% of pregnancies worldwide [134,135,136]. Despite the fact that risk factors for GDM are known to include being overweight or obese before pregnancy, advanced age, race, previous history of GDM, and family history of diabetes, the specific pathogenesis of GDM is not clear [137,138,139]. Increasing evidence suggest that several ncRNAs are dysregulated in the placenta of GDM patients and are associated with abnormalities in placental structure, metabolism, and function [23,140] (Table 3). Research on ncRNAs in the placenta of GDM patients will contribute to elucidating the pathogenesis of GDM, screening for GDM-related biomarkers, and identifying high-risk women with GDM as early as possible.

### 6.1. MiRNA and GDM

Recent studies have revealed that an increasing number of miRNAs play an essential role in the pathogenesis of GDM, yet how placenta-specific miRNAs and corresponding target genes are involved in the pathological process of GDM remains to be elucidated. Based on integrated miRNA and mRNA transcriptional profiling, Ding et al. identified 32 miRNAs and 281 mRNAs aberrantly expressed in the placenta of GDM patients, of which miR-138-5p is a critical gene [141]. Bioinformatics analysis showed that miR-138-5p was found to target the 3’-UTR of transducin β–like protein 1 (*TBL1X*), thereby inhibiting the proliferation and migration of trophoblast cells from participating in the development of GDM. Guan et al. [142] detected miR-21 expression levels in the placenta of 137 GDM patients and 158 healthy pregnant people by RT-qPCR, which showed that the expression of miR-21 was significantly downregulated in the GDM group. Further studies revealed that miR-21 can target peroxisome proliferator-activated receptor α (*PPAR-α*), which has a mutated binding site in the 3’-UTR that binds specifically to miR-21, thereby promoting trophoblast cell proliferation and migration. Similarly, miR-29b was found to be downregulated in the placenta of GDM patients and to target hypoxia-inducible factor 3A (*HIF3A*), which has two specific binding sites for miR-29b in the 3’-UTR, resulting in increased *HIF3A* expression, promoting trophoblast cell migration, and participating in GDM development [143].

Controlling inflammation and apoptosis in trophoblast cells is one of the critical aspects of the treatment of GDM. It was found that baicalein could achieve therapeutic effects in GDM by inhibiting the inflammation and apoptosis of trophoblast cells, which was mediated by miR-17-5p [150]. Meanwhile, they found that the expression of miR-17-5p was upregulated in the plasma and placenta of GDM patients. Zhang et al. [144] examined the expression levels of miR-30d-5p in the placentas of GDM patients and healthy control pregnant people, which showed that the expression levels of miR-30d-5p were significantly downregulated in the GDM group. Further studies revealed that miR-30d-5p promoted apoptosis and inhibited the proliferation, migration, invasion, and glucose uptake ability of trophoblast cells by negatively regulating the expression of Ras-related protein (*RAB8A*).

A link exists between the level of placental exosomes and placental dysfunction; therefore, exosomes play a key role in the study of the pathological process and treatment of GDM and represent an area of great interest [151,152]. The level of placenta-derived exosomes (PdE) was found to be higher in patients with GDM than in control pregnant people [153]. In addition, PdE levels were positively correlated with maternal body mass index (BMI), glucose concentration, and fetal weight, which means that PdE may be involved in maternal metabolic adaptation to pregnancy [153,154]. Zhang et al. [155] conducted a cross-sectional study to identify the expression levels of miRNAs in the placenta and circulating exosomes composed primarily of PdE in GDM patients, and 157 differentially expressed miRNAs were identified in GDM placental tissue, among which miR-125b was significantly downregulated and miR-144 was significantly upregulated in both the placenta and circulating exosomes. Further research revealed that these two miRNAs are mainly involved in the occurrence of GDM by affecting glucose metabolism [155]. Similarly, another study revealed that the placenta-derived exosomes miR-140-3p and miR-574-3p are significantly downregulated in patients with GDM and target *VEGF*, thereby promoting the proliferation, migration, and tube formation of umbilical vein endothelial cells [145]. The investigation of exosomal miRNAs will provide a better comprehension and exploration of the pathogenesis and development of GDM, enabling more effective diagnosis and new treatments.

### 6.2. LncRNA and GDM

lncRNAs are extensively involved in cell proliferation, migration, and apoptosis and are intimately related to the pathogenesis of GDM. Zhang et al. [146] identified the lncRNA maternally expressed gene 3 (*MEG3*) levels in the blood and placental villi tissue of GDM patients and control pregnant people by RT-qPCR; they found that *MEG3* expression levels were significantly upregulated in the GDM group, and *MEG3* overexpression may target miR-345-3p and reduce its level, thereby inhibiting trophoblast cell migration and invasion and inducing apoptosis. According to another study, lncRNA-*MALAT1* expression levels were increased in the placentas of GDM patients, while siRNA intervention could inhibit inflammation development and trophoblast cell proliferation, invasion, and migration by downregulating lncRNA-*MALAT1* expression, which may be achieved by regulating the TGF-β/ NF-κB signaling pathway [147].

Wang et al. [148] revealed that the expression level of LncRNA plasmacytoma variant translocation 1 (*PVT1*) in the placenta of GDM and PE patients was significantly lower than that of healthy placenta, which could significantly inhibit the invasion and proliferation of trophoblast cells. Further studies have indicated that *PVT1* positively regulates AKT phosphorylation and the expression of *GDPD3*, *ITGAV*, and *ITGB8*, thus, participating in GDM. Altered DNA methylation of *H19* and insulin-like growth factor 2 (*IGF2*)-imprinted genes is closely associated with fetal and placental development [156]. Su et al. [157] demonstrated that the expression levels of *IGF2* were significantly higher in cord blood and placental tissues of GDM patients, while the expression levels of *H19* were significantly lower. Further studies confirmed a strong association between IGF2/H19 methylation and intrauterine hyperglycemia-induced macrosomia.

Although the research on the molecular mechanisms and signaling pathways of lncRNAs involved in GDM has achieved some success, further research is needed to support the development of lncRNAs as clinically effective biomolecular markers for the treatment of GDM in the future.

### 6.3. CircRNA and GDM

Until now, few studies have been conducted on the relationship between placental circRNAs and GDM. Yan et al. [158] used next-generation sequencing (NGS) to identify 482 aberrantly expressed circRNAs in placental villus tissue from GDM patients, of which 227 were upregulated and 255 were downregulated. The GO and KEGG pathway revealed that these differentially expressed circRNAs were significantly enriched in pathways related to gluconeogenesis and lipid metabolism. Another study identified 46 differentially expressed circRNAs in GDM patients that were significantly enriched in the advanced glycation end-products receptor for the advanced glycation end-products (AGE-RAGE) signaling pathway, playing an important role in diabetic complications such as diabetic nephropathy and diabetic retinopathy [159,160].

Another study found that *circMAP3K4* was highly expressed in the placenta of GDM patients, and additionally, *circMAP3K4* could inhibit the insulin-PI3K/Akt signaling pathway via the miR-6795-5p/PTPN1 axis, which may be associated with GDM-associated insulin resistance [149]. The insulin-PI3K/Akt pathway promotes cellular glucose uptake and regulates cell growth, which is critical for insulin signaling [161]. In spite of these studies demonstrating the possible involvement of circRNAs in GDM, the specific mechanism of circRNA involvement in GDM has not been fully elucidated, and extensive experimental studies are needed to determine whether they can be used as potential biomarkers.

## 7. Macrosomia

The defining characteristic of macrosomia is a birth weight of over 4000 g, and the incidence of macrosomia has been increasing in recent decades [162,163,164]. The occurrence of macrosomia is an increased risk factor for maternal postpartum infections, and it is associated with several adverse perinatal outcomes, including prolonged labor and increased rates of cesarean delivery [165]. Compared to otherwise healthy infants, macrosomia leads to higher risks for childhood obesity, adult obesity, hypertension, diabetes, and other age-related diseases [166,167]. Existing research recognizes the critical role of regulating fetal growth played by placental ncRNAs (Table 4). The exploration of the correlation between placental ncRNAs and the birth outcome of macrosomia is beneficial to further understand the mechanism of macrosomia and to provide interventions and treatments.

### 7.1. MiRNA and Macrosomia

It is known that gestational diabetes (GDM) increases the risk of macrosomia [163,176,177,178]. Li et al. [168] compared miRNA profiles from the placentas of healthy and GDM pregnant women through microarray analysis and found that miR-508-3p was significantly upregulated in the GDM group and directly downregulated PIKfyve accordingly. PIKfyve is a phosphoinositide 3-phosphate-5-kinase that negatively regulates the epidermal growth factor receptor (*EGFR*). Therefore, increased expression of miR-508-3p in women with GDM is associated with inhibition of PIKfyve and abnormal activation of EGFR/PI3K/AKT signals, resulting in fetal overgrowth. At present, the prevention and control of the occurrence of macrosomia can be effectively achieved by strengthening the screening and management of GDM, but the cause of non-diabetic fetal macrosomia (NDFMS) remains to be studied.

Several studies have proven that the abnormal expression of miRNAs in the placenta is correlated with macrosomia. Guo et al. [169] identified miR-141-3p, a key miRNA with abnormal expression in the placenta of NDFMS, through preliminary screening of miRNA microarrays and further verification via quantitative RT-PCR (qRT-PCR). It was further proven that increased expression of miR-141-3p may regulate the proliferation of trophoblast cells in later pregnancy to participate in the onset and progression of NDFMS through an in vitro cellular model and a mouse pregnancy model.

Meanwhile, dysregulation of the miR-17-92 cluster in the placenta may cause various diseases, and previous studies have shown that miRNAs in the miR-17-92 cluster play an essential role in cell cycle migration, invasion, apoptosis, and proliferation. Li et al. [170] analyzed the expression levels of miRNAs in the miR-17-92 cluster between the placentas of healthy neonatal and macrosomia births, which found that miR-18a, miR-19a, miR-20a, miR-19b, and miR-92a were significantly increased in the macrosomia group compared to the healthy group, which may be due to the upregulation of miRNA-processing enzymes Drosha and Dicer. Additionally, they demonstrated that the miR-17-92 cluster targets *SMAD4* and *RB1* in trophoblast cells to attenuate cell apoptosis, promote cell proliferation, and accelerate cell entry into the S phase, which contributes to macrosomia development. According to another study, IGF2-derived miR-483-3p was found to be overexpressed in the placentas of macrosomia births and promoted trophoblast cell proliferation by downregulating its target gene RB1 inducible coiled-coil 1 (*RB1CC1*) [171].

### 7.2. LncRNA and Macrosomia

Song et al. [172] investigated the expression of lncRNAs in the placentas of average-birth-weight newborns and GDM macrosomia newborns, which showed that 2962 lncRNAs were upregulated and 1921 lncRNAs were downregulated in the placentas of macrosomia births compared to the healthy group. According to qRT-PCR, LncRNA *SNX17* showed a significant upregulation trend in the placentas of macrosomia births. Additionally, this lncRNA may promote the proliferation of trophoblast cells via the miR-517a/IGF-1 pathway and serve a function in the placental regulation of macrosomia.

Another study identified 2929 upregulated lncRNAs and 2127 downregulated lncRNAs in the placentas of macrosomia compared to average-birth-weight babies. It also demonstrated that lncRNAs have a significant differential expression in the placentas of macrosomia, which was probably associated with the development of NDFMS [179]. Lu et al. [173] identified 892 differentially expressed lncRNAs (763 upregulated and 129 downregulated) based on microarray analysis of lncRNAs in 48 NDFMS and 48 control placentas, which were further validated by selecting lncRNA *USP2-AS1* as a candidate lncRNA for subsequent experiments, and the downregulation of lncRNA *USP2-AS1* was found to possibly participate in NDFMS development by promoting trophoblast cell viability. Su et al. [157] found that lncRNA *H19* showed high methylation and low expression in the placental tissues of macrosomia with intrauterine hyperglycemia, and the methylation and ex-pression levels of lncRNA *H19* were significantly correlated with the birth weight of fetuses with intrauterine hyperglycemia. Until now, there have been several related studies on the association of lncRNAs with macrosomia, but the specific mechanism remains to be further investigated.

### 7.3. CircRNA and Macrosomia

Although fewer studies have been conducted on how circRNAs are involved in macrosomia, an association between the two was demonstrated. Wang et al. [174] demonstrated that circ-*SETD2* was significantly expressed in the placentas of macrosomia births compared with healthy donors via microarray assay. Additionally, bioinformatics analyses revealed that the expression of circ-*SETD2* was upregulated to strengthen the trophoblast cell proliferation and invasion, while miR-519a existed at the binding sites for both circ-*SETD2* and phosphate, and the tensin homolog was deleted on chromosome 10 (*PTEN*). Therefore, the circ-SETD2/miR-519a/PTEN axis regulates fetal weight by controlling trophoblast cell proliferation and invasion. These discoveries enhance our comprehension of the mechanisms of macrosomia, which can be explored in the future by focusing on the circ-SETD2/miR-519a/PTEN axis to facilitate the development of therapeutic strategies for this disease.

## 8. Low Birth Weight (LBW)

Low birth weight refers to full-term newborns with a birth weight lower than 2500 g, and it affects 15% to 20% of all newborns worldwide [180,181,182]. Fetal growth is influenced by the function of the placenta, which further affects the health of the newborn and increases the risk of disease in later life [183]. The human placenta expresses more than 600 miRNAs and has a specific miRNAs profile [184]. Thus, it is worth investigating the contribution of placental miRNAs and their crucial role in LBW, which is significant to understand the potential molecular mechanisms of LBW. Song et al. [175] identified a significant increase in placental miR-517a expression with LBW when compared to those with an average birth weight. Furthermore, it was found that overexpression of miR-517a significantly inhibited trophoblast cell invasion, suggesting that miR-517a may contribute to LBW through inhibition of trophoblast invasion. To date, there are few studies investigating the involvement of placental ncRNAs to regulate LBW, which leaves a need for further research in the future.

## 9. Birth Defects

Trisomy 21 (T21) was reported in recent studies as the main birth defect associated with abnormal expression of placental ncRNAs. T21 is the most common chromosomal aneuploidy, referred to as Down syndrome (DS), with an incidence of approximately 1:700 live births [185,186,187]. Individuals with T21 are at increased risk of developing a variety of diseases, including congenital heart defects, gastrointestinal abnormalities, leukemia, and neurodegenerative diseases [185,188,189]. To date, differential expression miRNAs in T21 are characterized as mainly originating from human chromosome 21, while a genome-wide comprehensive study of human genes and miRNAs may be critical for understanding the mechanisms underlying T21-related abnormalities.

Lim et al. [190] investigated the expression levels of genes and miRNAs in placental villus tissue of normal and T21 human fetuses and found significantly differential expression of 110 mRNAs (77 upregulated and 33 downregulated) and 34 miRNAs (16 upregulated and 18 downregulated). By predicting the functions and interactions between mRNAs and miRNAs using bioinformatics tools, they observed that there was a negative correlation between the expression levels of 8 miRNAs and 17 mRNAs in the T21 group, which includes 4 mRNAs with increased expression located on chromosome 21. In this study, they also identified the interaction network of three upregulated genes (*U2AF1*, *DYRK1A*, and *KSR1*), and four downregulated genes (*MRPL43*, *F2RL1*, *TICAM2*, and *MAP3K5*) in the placentas of T21, concluding that miRNA expression alteration may induce alterations in the levels of target genes (*DYRK1A*, *MAP3K5*) in T21, thus, playing a critical role in the pathogenesis of T21. Another study identified 12 differentially expressed miRNAs in T21 placenta, with a total of seven miRNAs confirmed to be upregulated (miR-99a, miR-542-5p, miR-10b, miR-125b, miR-615, *let-7c*, and miR-654), three of which (miR-99a, miR-125b, and *let-7c*) were located on chromosome 21 [191].

Modi et al. [192] investigated three miRNAs, miR-99a, miR-125b, and *let-7c*, for their expression patterns in chorionic tissue collected from full-term pregnancies with spontaneous rupture, and they demonstrated a potential role for these Chr-21 derived miRNAs in T21-related fetal membrane rupture and fetal membrane defects. Lim et al. [193] used microarray technology to analyze 1349 expression levels of miRNAs in whole blood and placenta samples from pregnant women with euploid or T21 fetuses and whole blood from non-pregnant women that identified 299 miRNAs that can be reasonably separated between the whole blood and placenta. Among the identified miRNAs, 150 miRNAs were upregulated and 149 were downregulated in the placentas, and most of the upregulated miRNAs were members of miR-498, miR-379, and miR-127 clusters. In addition, it was found that two miRNAs, miR-1973 and miR-3196, may regulate a total of 203 target genes participating in the brain, central nervous system, and neurological development and may be potential biomarkers for non-invasive testing for T21 in fetuses.

There is also a series of congenital diseases that were reported in recent years to be correlated closely with miRNAs in the placenta. For example, Radhakrishna et al. [194] investigated the levels of miRNAs in the placentas of ventricular septal defect (VSD) births, which revealed that miR-191, miR-548F1, miR-148A, miR-423, miR-92B, miR-611, miR-2110, and miR-548H4 showed significant changes and that these eight miRNAs could be used as potential biomarkers for the detection of VSD. Although such similar research is just beginning, it is worth waiting for a clearer recognition of the relationship between placental ncRNAs and congenital anomalies.

## 10. Conclusions

Recently, numerous studies have emerged offering broad discussions about placental ncRNAs and APOs. Most of the studies mainly evaluated the specific miRNAs and lncRNAs associated with APOs, while the field of circRNAs is still lacking. Notably, miRNAs, lncRNAs, and circRNAs may be involved in gene regulation by two or three of them together in the development of disease. Accordingly, we expected more comprehensive studies on circRNAs and APOs that could also be conducted to bridge the current gaps in the research. Current studies mainly analyze the relationship between the differential expression of ncRNAs and APOs by high-throughput sequencing, while future work is needed to fully understand the implications of the specific mechanisms related to disease occurrence and development. In summary, the study of ncRNAs and gene regulatory networks related to APOs is beneficial for further understanding the pathogenesis of APOs, and this may also provide new ideas for early diagnosis, prevention, and treatment.

## Figures and Tables

**Figure 1 ijms-24-05030-f001:**
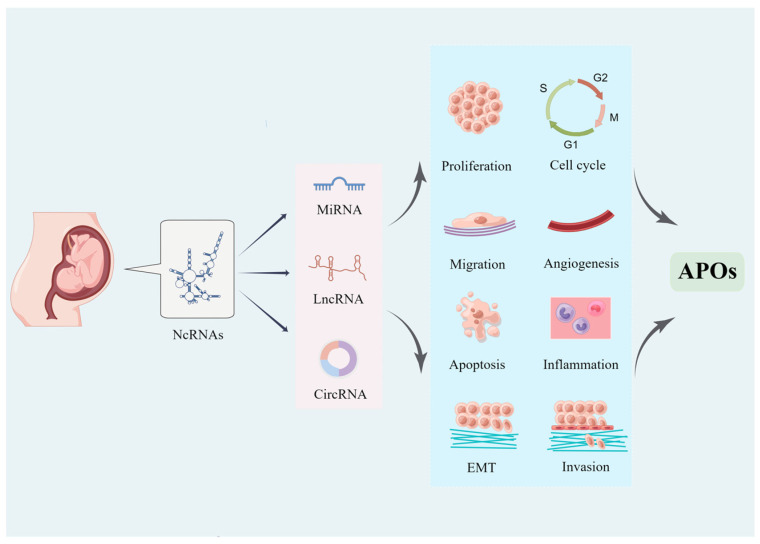
Placental ncRNAs participate in adverse pregnancy outcomes.

**Figure 2 ijms-24-05030-f002:**
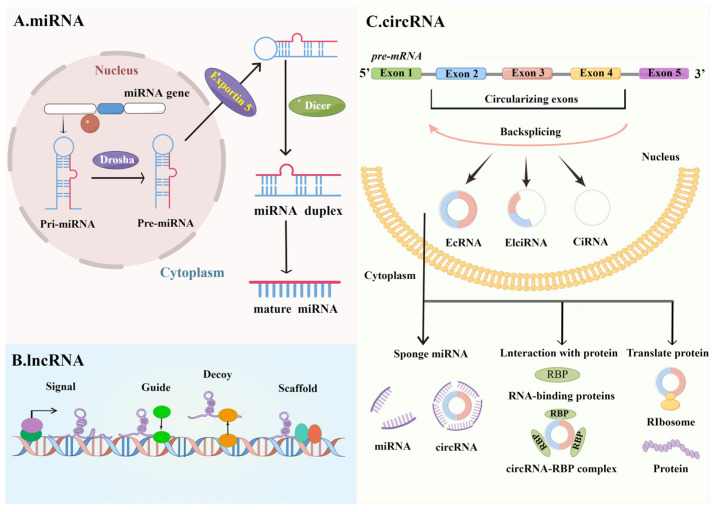
Biogenesis and function of main ncRNAs, including miRNAs, lncRNAs, and circRNAs.

**Table 1 ijms-24-05030-t001:** The role of ncRNAs in preterm birth and miscarriage.

Pregnancy Outcomes	RNA Categories	RNA Names	Expression	Function	Reference
Preterm birth	MiRNA	MiR-519c	Downregulation	Anti-inflammatory action	[61]
	LncRNA	*SNHG29*	Upregulation	Activates P53/P21 signals and triggers cellular senescence	[62]
Miscarriage	MiRNA	MiR-4497	Upregulation	Downregulates *SP1* mRNA and induces trophoblast cell apoptosis	[63]
		MiR-135a-5p	Upregulation	Targets *PTPN1* and inhibits proliferation, invasion, and migration of trophoblast cells	[64]
		MiR-27a-3p	Upregulation	Downregulates *USP25* and regulates the process of EMT, migration, and invasion of trophoblast cells	[65]
	LncRNA	*H19*	Downregulation	Inhibits let-7and upregulates *ITGB3* expression	[66]
		*SNHG7-1*	Downregulation	Regulates the proliferation and invasion of trophoblast cells by targeting miR-34a	[67]
		*MEG8*	Upregulation	Inhibits cell proliferation and invasion of trophoblast cells	[68]
	CircRNA	Circ-*ZUFSP*	Upregulation	Regulates migration and invasion of trophoblast cells via the CIRC-ZUFSP/miR-203/STOX1 pathway	[69]
		*CircPUM1*	Downregulation	Promotes trophoblast cell processes and anti-inflammatory effects via the miR-30a-5p/JunB axis	[70]
		*CircFOXP1*	Downregulation	Regulates the function of trophoblast cells via the miR-143-3p/S100A11 pathway	[71]
		Circ-*DURSA*	Downregulation	Regulates trophoblast cell apoptosis via miR-760-HIST1H2BE axis	[72]

**Table 2 ijms-24-05030-t002:** The role of ncRNAs in pre-eclampsia.

Pregnancy Outcomes	RNA Categories	RNA Names	Expression	Function	Reference
Pre-eclampsia	MiRNA	MiR-513c-5p	Upregulation	Promotes trophoblast cell apoptosis and inhibits cell proliferation, invasion, and migration by targeting *LRP6*	[106]
		MiR-21	Upregulation	Targets *FOXM1* and inhibits trophoblast cell proliferation	[107]
		MiR-195	Downregulation	Regulates trophoblast cell proliferation, apoptosis, and angiogenesis	[108]
		MiR-145-5p	Downregulation	Inhibits trophoblast cell proliferation and invasion	[109]
		MiR-30a-3p	Upregulation	Regulates trophoblast cell invasion and apoptosis	[110]
		MiR-548c-5p	Downregulation	Anti-inflammatory action	[111]
	LncRNA	*DLX6-AS1*	Upregulation	Regulates the proliferation, invasion, and angiogenesis of trophoblast cells	[112]
		*XIST*	Upregulation	Inhibits trophoblast cell proliferation and invasion and induces cell apoptosis	[113]
		*SNHG16*	Downregulation	Inhibits trophoblast cell proliferation, migration, and invasion and induces apoptosis	[114]
		*SNHG22*	Downregulation	Promotes migration and invasion of trophoblast cells via miR-128-3p/PCDH11X axis	[115]
		*TUG1*	Downregulation	Promotes trophoblast cell proliferation, invasion, migration, and angiogenesis	[116,117]
		*LINC00534*	Upregulation	Promotes trophoblast cell apoptosis and inhibits proliferation and migration via miR-494-3p/PTEN axis	[118]
		*BCYRN1*	Downregulation	Inhibits cell viability, migration, invasion, and tube forming abilities	[119]
	CircRNA	*CircSFXN*	Upregulation	Inhibits trophoblast cell invasion and angiogenesis	[120]
		*CircZDHHC20*	Upregulation	Inhibits trophoblast cell proliferation, migration, and invasion	[121]
		*CircPAPPA*	Downregulation	Regulates trophoblast cell proliferation, migration, apoptosis, and invasion	[122,123]
		Has_circ_0008726	Upregulation	Inhibits trophoblast cell proliferation, migration, invasion, and EMT	[124,125]

**Table 3 ijms-24-05030-t003:** The role of ncRNAs in gestational diabetes mellitus.

Pregnancy Outcomes	RNA Categories	RNA Names	Expression	Function	Reference
gestational diabetes mellitus	MiRNA	MiR-138-5p	Downregulation	Inhibits proliferation and migration of trophoblast cells	[141]
		MiR-21	Downregulation	Promotes trophoblast cell proliferation and migration	[142]
		MiR-29b	Downregulation	Promotes trophoblast cell migration	[143]
		MiR-30d-5p	Downregulation	Promotes apoptosis and inhibits the proliferation, migration, invasion, and glucose uptake ability of trophoblast cells	[144]
		MiR-140-3p	Downregulation	Promotes the proliferation, migration, and tube formation of umbilical vein endothelial cells	[145]
		MiR-574-3p	Downregulation	Promotes the proliferation, migration, and tube formation of umbilical vein endothelial cells	[145]
	LncRNA	*MEG3*	Upregulation	Inhibits trophoblast cell migration and invasion and induces apoptosis	[146]
		lncRNA-*MALAT1*	Upregulation	Inhibits trophoblast cell proliferation, invasion, and migration and anti-inflammatory action	[147]
		*PVT1*	Downregulation	Inhibits invasion and proliferation of trophoblast cells	[148]
	CircRNA	*CircMAP3K4*	Upregulation	Inhibits the insulin-PI3K/Akt signaling pathway	[149]

**Table 4 ijms-24-05030-t004:** The role of ncRNAs in macrosomia and low birth weight.

Pregnancy Outcomes	RNA Categories	RNA Names	Expression	Function	Reference
Macrosomia	MiRNA	MiR-508-3p	Upregulation	Inhibits PIKfyve and abnormal activation of EGFR/PI3K/AKT signals	[168]
		MiR-141-3p	Upregulation	Stimulates placental cell proliferation	[169]
		MiR-18a/19a/20a/92a/19b	Upregulation	Promotes cell proliferation, attenuates cell apoptosis, and accelerates cells entering S phase by targeting *SMAD4* and *RB1* in trophoblast cells.	[170]
		MiR-483-3p	Upregulation	Targets *RB1CC1* and promotes trophoblast cell proliferation	[171]
	LncRNA	LncRNA-*SNX17*	Upregulation	Promotes trophoblast cell proliferation through miR-517a/IGF-1 pathway	[172]
		*USP2-AS1*	Downregulation	Promotes trophoblast cell viability	[173]
	CircRNA	Circ-*SETD2*	Upregulation	Regulates trophoblast cell proliferation via circ-SETD2/miR-519a/PTEN axis	[174]
Low birth weight	MiRNA	MiR-517a	Upregulation	Inhibits trophoblast cell invasion	[175]

## Data Availability

The figure was created by Figdraw (http://www.figdraw.com, accessed on 29 November 2022).

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
