# Peer review of "The Role of Placental Non-Coding RNAs in Adverse Pregnancy Outcomes"

_ijms, 2023, doi:10.3390/ijms24055030_

Round 1

Reviewer 1 Report

The manuscript summarizes current knowledge about the role of different non coding RNAs in adverse pregnancy outcome. This is an interesting review that summarizes the current knowledge from the field and deserves to be published.

Minor comments

I suggest paragraphs not to start with abbreviation.

Reviewer 2 Report

In the present work, Ren, Jin and Zhu offer an attractive field of study in the field of adverse outcomes of pregnancy (APOs), exploring the updated knowledge about the role of non-coding RNAs (ncRNAs) in these conditions. In general terms, the article is interesting and covers a promising area of research with multiple translational applications. However, there are some remarkable issues that should be addressed to increase the quality and value of the manuscript

Major points

1) I would strongly recommend to include additional information about the role of miRNAs in two major adverse pregnancy outcomes: Gestational diabetes mellitus (GDM) and hypertensive disorders of pregnancy, particularly pre-eclampsia (PE). Despite the former is partially emphasized in relation with macrosomia, due to its frequency and detrimental effects on the foetus and mother should be explored in another section. On the other hand, the latter is perhaps one of the most challenging APOs with unclear pathophysiology and clinical approaches and many short-term and long-term complications related. I am sure that ncRNAs can shed light into these complex entities, giving an extra value to the present manuscript 2) A visual figure or scheme in section 2 representing the main features, function and biosynthesis of the different types of ncRNAs could aid to the readers to better understand the manuscript. 

Minor comments

Line 36  maternal and fetalà mother and foetus

Line 41 Please, redefine the abbreviature of ncRNAs in the introduction. 

Line 375. More recently, more literature has emerged…. It sounds a little redundant, please rewrite.

Reviewer 3 Report

Dear Editor,

The authors have chosen a very good and interesting topic.They have also used good literature search to elucidate their findings. But as per my understanding, the authors should take following points into consideration to make the article more comprehensive and lucid.

1. The writing style needs significant improvement. Please check for grammatical and contextual errors. Rephrasing of sentences would improve the overall quality of the manuscript.

2. SNPs in many miRNAs/ncRNAs have been shown to be involved in the pathogenesis of miscarriages. Author should discuss this aspect also with recent references (Srivastava P, Bamba C, Chopra S, Mandal K. Role of miRNA polymorphism in recurrent pregnancy loss: a systematic review and meta-analysis. Biomark Med. 2022;16(2):101-115. doi:10.2217/bmm-2021-0568).

3. Many sentences used in the text are little bit longer. Therefore, the authors should go through the text again and try to succinct the long sentences.

4. Exosomal miRNAs are also emerging as a therapeutic aspect for pregnancy-related issues. Elaborate on this point.

Round 2

Reviewer 2 Report

The authors have made significant changes. However, some aspects need to be improved:

-The authors should consider improving the title of the manuscript, with more specific aspects.

-The authors must include aspects related to the effects of lipidomics in the placenta. One reference that authors should include is doi: 10.7150/ijms.49236.

-Authors should include references to systemic vascular diseases such as doi: 10.3390/cells11030568.

-The role of extracellular vesicles should be included with the reference doi: 10.3389/fcell.2022.1060850.

-Authors should improve the use of English grammar with experts in the area.
